# Supramolecular Assemblies in Pb(II) Complexes with Hydrazido-Based Ligands

**Ghodrat Mahmoudi [1],\***, **Saikat Kumar Seth [2]**, **Fedor I. Zubkov [3]**, **Elena López-Torres [4]**, **Alessia Bacchi [5]**, **Vladimir Stilinović [6] and Antonio Frontera [7],\***

[1] Department of Chemistry, Faculty of Science, University of Maragheh, Maragheh P.O. Box 55181-83111, Iran
[2] Department of Physics, Jadavpur University, Kolkata 700032, India; saikatim@yahoo.co.in
[3] Organic Chemistry Department, Faculty of Science, Peoples' Friendship University of Russia (RUDN University), 6 Miklukho-Maklaya St., Moscow 117198, Russia; fzubkov@sci.pfu.edu.ru
[4] Departamento de Química Inorgánica, Facultad de Ciencias, Módulo 07, Universidad Autónoma de Madrid, Ctra. de Colmenar Viejo, Km 15, 28049 Madrid, Spain; elena.lopez@uam.es
[5] Dipartimento di Scienze Chimiche, della Vita e della Sostenibilità Ambientale, Università di Parma, Viale delle Scienze 11A, 43124 Parma, Italy; alessia.bacchi@unipr.it
[6] Department of Chemistry, Faculty of Science, University of Zagreb, Horvatovac 102a, HR-10000 Zagreb, Croatia; vladimir_stilinovic@yahoo.com
[7] Departament de Química, Universitat de les Illes Balears, Ctra. de Valldemossa km 7.5, 07122 Palma de Mallorca (Baleares), Spain
**\*** Correspondence: ghodratmahmoudi@gmail.com (G.M.); toni.frontera@uib.es (A.F.)

**Abstract:** Herein, we describe the synthesis and single crystal X-ray diffraction characterization of several Pb(II) complexes using Schiff base hydrazido-based ligands and different counterions ($NO_3^-$, $I^-$ and $ClO_4$). In the three complexes reported in this work, the lead(II) metal exhibits a high coordination number (n > 8) and thus it is apparently not involved in tetrel bonding interactions. Moreover, the aromatic ligands participate in noncovalent interactions that play an important role in the formation of several supramolecular assemblies in the solid state of the three Pb(II) complexes. These assemblies have been analyzed by means of Hirshfeld surface analysis and DFT calculations.

**Keywords:** lead(II) coordination; crystal engineering; noncovalent interactions; π-stacking; hydrogen bonding

## 1. Introduction

The large radius of lead(II) is responsible of its versatile coordination chemistry and its coordination number usually ranges from four to nine. Consequently, it has been used for the preparation of new hybrid inorganic-organic compounds, polymers and complexes [1–4]. It is well-known that the utilization of lead(II) has negative implications related to environmental issues and health; however, it used for the synthesis of materials with interesting properties like semiconductors, NLO (non-linear optical) and ferroelectric materials [5–8]. In fact, nowadays the technical, economic and social relevance of Pb is beyond all doubt [9].

The study of lead coordination compounds is of general interest because the development of ligands capable to trap lead(II) from drinking water, paints or even the human body is necessary [9–15]. It is well-known that the $6s^2$ electron pair of Pb(II) can either remain inert, or be stereochemically active as widely discussed in the literature [16–23]. Tetrel bond interactions with hemi-directionally coordinated lead have been studied in the literature [24–26] and are relevant in supramolecular chemistry [27–29] and metal organic frameworks (MOF) based on lead(II) [30].

In the present work, three nicotinhydrazido and picolinhydrazido $N_2O$ and $N_2O_2$ donor Schiff bases, (*E*)-*N*′-(2-pyridinylmethylene)nicotinohydrazide ($L^1$), (*E*)-*N*′-(phenyl(pyridin-2-yl)methylene)picolinohydrazide ($L^2$) and *N*′,*N*′′′-((1*E*,2*E*)-1,2-diphenylethane-1,2-diylidene)di(picolinohydrazide) ($L^3$) have been used to prepare and X-ray characterize one dinuclear and two mononuclear lead(II) complexes (see Figure 1): $[Pb_2(L^1)_4(NO_3)_2][ClO_4]_2$ (**1**), $[Pb(L^2)_2I_2]$ (**2**), and $[Pb(L^3)_2(H_2O)][ClO_4]_2$ (**3**). Moreover, supramolecular assemblies have been studied using density functional theory (DFT) calculations and Hirshfeld surface analyses. In these complexes the conventional coordination number is n ≥ 8 and consequently holodirected coordination mode is observed.

**Figure 1.** Schematic representation of ligands and Pb(II) coordination compounds (**1**–**3**) used in this work and their synthetic route.

## 2. Materials and Methods

### 2.1. Materials

Ligands were synthesized following the protocol previously published by us [24].

### 2.2. Synthesis of the Crystals 1–3

In order to grow single-crystals, we used a special glassware that consists in a branched tube designed by us [31] for doing the reaction and crystallization in a single step (see reference [31] for a detailed explanation). The detailed synthesis of **1** is explained here. Complexes **2** and **3** were synthesized using the same synthetic protocol, utilizing Pb(NO₃)₂ and $L^1$ for **2** and PbI₂ and HL² for **3**.

Synthesis of complex **1**: Pb(NO₃)₂, $L^1$ and NaClO₄ (0.164 g, 0.500 mmol; 0.113 g, 0.500 mmol; and 0.061 g, 0.500 mmol; respectively) were introduced in the special glassware (main arm of the tube). Both arms of the branched tube were filled with 15 ml of MeOH and the main arm introduced in an

oil bath at 60 °C and the branched arm was preserved at room temperature. In the branched arm, crystals of **1** were isolated after 6 days. The crystals were filtered off and, subsequently, washed with $CH_3COCH_3$ and $Et_2O$, and finally dried in air.

$[Pb_2(L^1)_4(NO_3)_2][ClO_4]_2$ (**1**): Isolated yield was 72%. Anal. calcd. (found) for $C_{48}H_{40}Cl_2N_{18}O_{18}Pb_2$; C, 35.11 (35.01); H, 2.46 (2.51); N, 15.350 (15.60)%. IR (cm$^{-1}$) selected bands: $\tilde{v}$ = CH (bending, out-of-plane): 667 (m) and 779 (m); Cl-O (stretching): 1101 (m); NO (stretching): 1378 (m); CC (stretching): 1460 (m); C=N (stretching): 1499 and 1579 (m); C=O (stretching) 1670; CH (stretching): 3007 (w).

$[Pb(L^2)_2I_2]$ (**2**): Isolated yield was 87%. Anal. calcd. (found) for $C_{36}H_{28}I_2N_8O_2Pb$; C, 40.57 (40.71); H, 2.65 (2.75); N, 10.51 (10.15)%. IR (cm$^{-1}$) selected bands: $\tilde{v}$ = CH (bending, out-of-plane): 701 (m) and 784 (m); CC (stretching): 1428 (m); C=N (stretching): 1493 and 1589 (m); C-O (stretching) 1676; CH (stretching): 2923 (w); NH: 3284 (m).

$[Pb(L^3)_2(H_2O)][(ClO_4)]_2$ (**3**): Isolated yield was 67%. Anal. calcd. (found) for $C_{52}H_{42}N_{12}O_{13}PbCl_2$; C, 47.28 (47.67); H, 3.20 (3.24); N, 12.72 (12.51)%. IR (cm$^{-1}$) selected bands: $\tilde{v}$ = CH (bending, out-of-plane): 688 (m) and 769 (m); Cl-O (stretching): 1100 (m); CC (stretching): 1435 (m); C=N (stretching): 1567 and 1586 (m); C=O (stretching) 1672; CH (stretching): 2926 (w); OH: 3445 (m).

### 2.3. Characterization

*X-ray Diffraction analyses.* Single crystal X-ray diffraction of compounds (**1–3**) were collected on Bruker APEX-II CCD diffractometer at 293(2) K with MoKα radiation (λ = 0.71073 Å). The Bruker SAINT [32] program was used for data reduction and by using multi-scan method [33], the absorption correction was applied. The structure of (**1–3**) were solved by using the program SHELXS-14 [34] and refined by SHELXL-18 [35]. All hydrogen atoms were located at their geometrically perfect positions and an isotropic refinement was employed. All calculations were performed using the programs WinGX system V2014.1 [36] and PLATON [37]. The details of the crystal data and structure refinement factors are included in Table 1. (CCDC 1919248–1919250) contain the supplementary crystallographic data of compounds (**1–3**) respectively.

### 2.4. Hirshfeld Surface Analysis

Hirshfeld surface (H-S) [38–40] is generated based on electron distribution of investigating molecule. H-S is distinctive [41] for every crystal structure. The normalized contact distance ($d_{norm}$) calculated by Equation (1) is used to locate both inner and outer intermolecular interactions simultaneously on a single Hirshfeld surface [38]. Where, $d_e$ and $d_i$ are the distances from the point to the nearest nucleus external and internal to the surface respectively; $r_i^{vdw}$ and $r_e^{vdw}$ are the internal and external van der Walls (vdW) radii of the two atoms to the surface respectively. The 2D fingerprint plot displays the summary of available contacts in the crystal [42–46] that are generated by binning ($d_e$, $d_i$) pairs in the interval of 0.01 Å. The 2D histogram was produced as a function of the fraction of surface points in that bin (essentially a pixel) that varies from few points to many points i.e., from blue through green to red. The Hirshfeld surfaces are plotted with $d_{norm}$ using red-white-blue color scheme, where red color indicates shorter contacts, white shows the contacts around the *vdW* separation, and blue designate longer contacts. Additionally, another colored property of the surface named "shape-index" has been discussed in this manuscript. Shape-index is a measure of "which shape", and extremely sensitive in low curvature areas. The Hirshfeld surface calculation was performed using the program *Crystal Explorer 3.1.* [47].

$$d_{norm} = \frac{d_i - r_i^{vdw}}{r_i^{vdw}} + \frac{d_e - r_e^{vdw}}{r_e^{vdw}} \tag{1}$$

## 2.5. Computational Details

For the calculations we have used the X-ray coordinates where only the position of the H-atoms were optimized using DFT calculations at the PBE0 [48]-D [49]/def2-SVP [50] level of theory. Gaussian-09 [51] program package was used for the calculations. The def2-svp basis set that uses ECPs (effective core potentials) [52] for Pb and I atoms.

**Table 1.** Crystallographic data and structure refinement factors of (**1**–**3**).

| Structure | (1) | (2) | (3) |
|---|---|---|---|
| Empirical formula | $C_{48}H_{40}Cl_2N_{18}O_{18}Pb_2$ | $C_{36}H_{28}I_2N_8O_2Pb$ | $C_{52}H_{41}Cl_2N_{12}O_{13}Pb$ |
| Formula Weight | 1642.26 | 1065.65 | 1320.06 |
| Temperature (K) | 296(2) | 296(2) | 293(2) |
| Wavelength (Å) | 0.71073 | 0.71073 | 0.71073 |
| Crystal system | Monoclinic | Triclinic | Monoclinic |
| space group | P 2₁/c | P-1 | C2/c |
| a, b, c (Å) | 10.5189(8), 20.6580(2), 13.455(1) | 8.8711(1), 10.4600(2), 10.5452(2) | 16.2884(9), 15.4978(9) 21.409(2) |
| α, β, γ (°) | 90, 92.218(1), 90 | 79.134(1), 77.219(1), 74.3900(10) | 90, 100.016(6), 90 |
| Volume (Å³) | 2921.6(3) | 910.36(3) | 5321.9(6) |
| Z/Density (calc.) (Mg/m³) | 2/1.867 | 1/1.944 | 4/1.648 |
| Absorption coefficient (mm⁻¹) | 5.933 | 6.373 | 3.347 |
| F(000) | 1592 | 504 | 2628 |
| Crystal size (mm³) | 0.26 × 0.20 × 0.17 | 0.20 × 0.14 × 0.02 | 0.23 × 0.11 × 0.07 |
| θ range for data collection | 1.807 to 26.998 | 2.00 to 26.33 | 3.866 to 26.999 |
| Limiting indices | −13 ≤ h ≤ 13 −26 ≤ k ≤ 26 17 ≤ l ≤ 17 | −11 ≤ h ≤ 11, −13 ≤ k ≤ 13, −13 ≤ l ≤ 13 | −20 ≤ h ≤ 20 −19 ≤ k ≤ 19 −27 ≤ l ≤ 27 |
| Reflections collected/unique | 36943/6377 | 13426/3639 | 19102/4472 |
| Completeness to θ (%) | 100.0 | 98.3 | 99.5 |
| Absorption correction | Semi-empirical from equivalents | Semi-empirical from equivalents | Semi-empirical from equivalents |
| Max. and min. transmission | 0.382 and 0.265 | 0.890 and 0.352 | 1.000 and 0.785 |
| Refinement method | Full-matrix least-squares on F² | Full-matrix least-squares on F² | Full-matrix least-squares on F² |
| Data/parameters | 6377/393 | 3639/223 | 5776/363 |
| Goodness-of - fit on F² | 1.034 | 1.021 | 0.968 |
| Final R indices [I > 2σ(I)] | $R_1 = 0.0343, wR_2 = 0.0763$ | $R_1 = 0.0288, wR_2 = 0.0506$ | $R_1 = 0.0360, wR_2 = 0.0821$ |
| R indices (all data) | $R_1 = 0.0542, wR_2 = 0.0836$ | $R_1 = 0.0403, wR_2 = 0.0540$ | $R_1 = 0.0518, wR_2 = 0.0847$ |
| Largest diff. peak and hole (e.Å⁻³) | 0.986 and −0.702 | 0.570 and −0.779 | 1.188 and −0.685 |

$R_1 = \sum\|F_o|-|F_c\|/\sum|F_o|$, $wR_2 = [\sum\{(F_o{}^2 - F_c{}^2)^2\}/\sum\{w(F_o{}^2)^2\}]^{1/2}$, $w = 1/\{\sigma^2(F_o{}^2) + (aP)^2 + bP\}$, where a = 0.0371 and b = 0.9414 for (**1**); a = 0.0188 and b = 0.2899 for (**2**) and a = 0.0486 and b = 0.0000 for (**3**). $P = (F_o{}^2 + 2F_c{}^2)/3$ for all structures.

## 3. Results and Discussion

### 3.1. Structural Analysis

The molecular view of compound (**1**) is included in Figure 2 with partial atom-numbering. Four nitrogen and two oxygen atoms from hydrazide and 2-pyridinyl groups of two ligands form a distorted $PbN_4O_2$ chromophore. The oxygen atom from the nitrate anion occupies the seventh coordination site with a distance of 2.844(2)Å (Table 2). The N(4A) atom of the ligand occupies eighth coordination site. The long Pb-N(4A) and Pb–O3 coordination bonds (>2.8 Å) can be also described as strong (noncovalent) tetrel bonds instead of coordination bonds. It fact, these distances are approximately in the middle of $\Sigma R_{cov}$ (sum of covalent radii: 2.12 for Pb + O and 2.17 Pb + N) and $\Sigma R_{vdw}$ (sum of van der Waals radii: 3.54 for Pb + O and 3.57 Pb + N). Based on this simple

geometric criterion, both interactions can be either described as tetrel bonds or weak coordination bonds. Therefore, in case of compound **1**, where the Pb–N(4A) and Pb–O3 distances are significantly longer than $\Sigma R_{cov}$, the structure can be also defined as a mononuclear $[Pb(L1)_2][NO_3]\cdot[ClO_4]$ hemidirected (n = 6) complex self-assembled by two strong Pb$\cdots$N4 tetrel bonds.

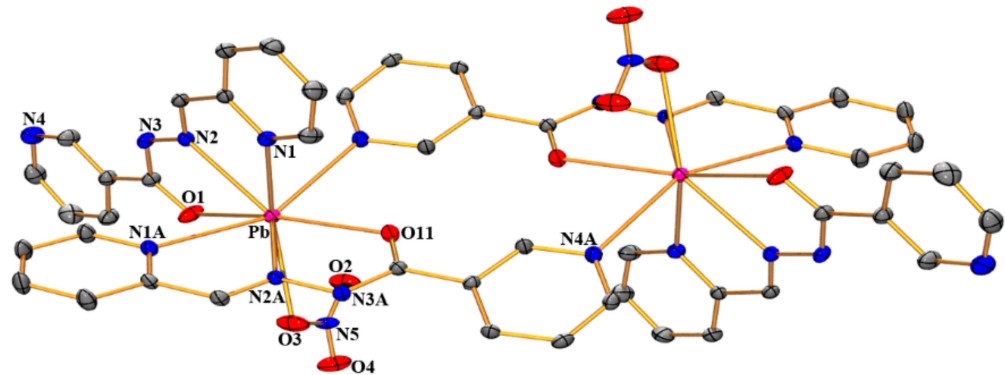

**Figure 2.** Molecular view of compound (**1**) with partial atom numbering. The unlabeled counterpart is produced through the symmetry (1 − x, −y, 1−z). Perchlorate anion is not included for clarity. Ellipsoids are drawn at 30% probability.

**Table 2.** Selected bond lengths (Å) and bond angles (°).

| | *Compound (1)* | | |
|---|---|---|---|
| **Pb-N(1)** | **2.614(4)** | **Pb-N(2A)** | **2.723(3)** |
| Pb-N(2) | 2.646(4) | Pb-N(1A) | 2.744(4) |
| Pb-O(1) | 2.775(2) | Pb-O(3) | 2.844(2) |
| Pb-O(11) | 2.648(3) | Pb-N(4A) | 2.816(2) |
| N(1)-Pb-N(2) | 62.08(13) | O(11)-Pb-N(2A) | 57.92(11) |
| N(1)-Pb-O(11) | 79.64(13) | N(1)-Pb-N(1A) | 84.10(13) |
| N(2)-Pb-O(11) | 138.55(13) | N(2)-Pb-N(1A) | 76.60(12) |
| N(1)-Pb-N(2A) | 82.70(12) | O(11)-Pb-N(1A) | 116.63(12) |
| N(2)-Pb-N(2A) | 126.17(12) | N(2A)-Pb-N(1A) | 59.47(12) |
| | *Compound (2)* | | |
| Pb(1)-O(1) | 2.710(3) | Pb(1)-N(1) | 2.833(2) |
| Pb(1)-I(1) | 3.188(3) | Pb(1)-N(3) | 2.917(2) |
| O(1)-Pb(1)-O(1) * | 180.00(8) | O(1)-Pb(1)-N(1) | 57.84(7) |
| O(1)-Pb(1)-I(1) | 86.61(7) | N(1)-Pb(1)-I(1) | 99.97(8) |
| O(1)-Pb(1)-I(1) * | 93.39(7) | N(I)-Pb(1)-I(1) * | 80.03(11) |
| I(1)-Pb(1)-I(1) * | 180.00(7) | N(1)-Pb(1)-N(1) * | 180.00(12) |
| | *Compound (3)* | | |
| Pb(1)-O(1) | 2.581(3) | Pb(1)-N(4) | 2.701(3) |
| Pb(1)-N(3) | 2.675(3) | Pb(1)-O(7) | 2.737(7) |
| O(1)-Pb(1)-N(3) | 60.39(9) | O(1)-Pb(1)-O(7) | 142.78(7) |
| O(1)-Pb(1)-N(4) | 115.35(9) | N(3)-Pb(1)-O(7) | 117.30(7) |
| N(3)-Pb(1)-N(4) | 57.86(9) | N(4)-Pb(1)-O(7) | 84.17(6) |

In addition of these tetrel bonds, we have also explored the weak interactions that played crucial role for the construction of supramolecular networks in solid-state. In (**1**), two oxygen atoms of the nitrate and perchlorate anion acts as donor to the carbon atoms in the molecule at (x, 1/2 − y, −1/2 + z) and (1 − x, −1/2 + y, 3/2 − z) respectively (Table 3); thus generating a supramolecular network in (011) plane (Figure S1, see Supplementary Materials). Again, the nitrogen atom N(3A) binds nitrate oxygen atom O(3) at (−x, −y, 1 − z); consequently creating a centrosymmetric dimeric ring $R_2^2(10)$ centered at (0 0 $\frac{1}{2}$). The molecules propagate along (100) direction due to the self-complementary nature of the N-H$\cdots$O bond that generates the dimeric ring. The parallel chains are interlinked through another N-H$\cdots$O bond to produce a layered network in (101) plane (Figure 3).

**Table 3.** Geometric parameters of hydrogen bonds (Å, °).

| D–H···A | D–H | H···A | D···A | D–H···A | Symmetry Operation |
|---|---|---|---|---|---|
| *Compound (**1**)* | | | | | |
| N3A–H3B···O3 | 0.90 | 2.10 | 2.975(6) | 164 | −x, −y, 1 − z |
| N3A–H3B···O4 | 0.90 | 2.58 | 3.309(7) | 139 | −x, −y, 1 − z |
| N3–H3N···O2 | 0.90 | 2.15 | 2.970(6) | 151 | x, 1/2 − y, −1/2 + z |
| C3–H3···O8 | 0.93 | 2.29 | 3.068(10) | 141 | x, y, −1 + z |
| C4–H4···O6 | 0.93 | 2.50 | 3.299(12) | 144 | 1 − x, −y, 1 − z |
| C5–H5···O11 | 0.93 | 2.58 | 3.239(8) | 128 | - |
| C6–H6···O2 | 0.93 | 2.53 | 3.178(6) | 127 | x, 1/2 − y, −1/2 + z |
| C9–H9···O6 | 0.93 | 2.51 | 3.247(10) | 136 | x, 1/2 − y, −1/2 + z |
| C9A–H9A···O3 | 0.93 | 2.41 | 3.317(7) | 165 | −x, −y, 1 − z |
| C11A–H11A···O8 | 0.93 | 2.51 | 3.300(11) | 143 | 1 − x, −1/2 + y, 3/2 − z |
| C12A–H12A···O11 | 0.93 | 2.33 | 3.072(6) | 136 | 1 − x, −y, 1 − z |
| *Compound (**2**)* | | | | | |
| N2–H2···N4 | 0.90 | 2.24 | 2.655(5) | 108 | - |
| C13–H13···O1 | 0.93 | 2.37 | 2.982(6) | 123 | −x, 2 − y, 1 − z |
| *Compound (**3**)* | | | | | |
| N2–H1N···N1 | 0.86 | 2.33 | 2.673(4) | 104 | - |
| O7–H1W···O5 | 0.82 | 2.41 | 3.214(6) | 169 | - |
| N5–H2N···N6 | 0.86 | 2.25 | 2.625(5) | 107 | - |
| C25–H25···O6 | 0.93 | 2.57 | 3.424(6) | 153 | x, 1 − y, 1/2 + z |

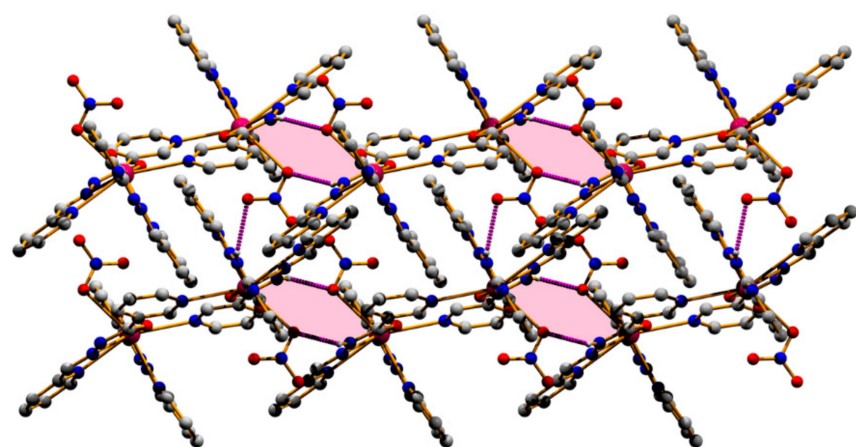

**Figure 3.** Self-assembled layered framework in (**1**) generated by strong N-H···O H-bonds in (101) plane.

Nevertheless, oxygen atom O(5) of perchlorate anion is leaning towards the π-cloud of the pyridine ring having separation distance of 3.498(6)Å (Table 4) at (1 − x, −y, 1 − z), suggesting anion···π interaction [53–55]. Again, the π–π stacking interactions (Table 5) between two pyridine rings of adjacent chains are optimized. The pyridine rings at (x, y, z) and (x, 1/2 − y, −1/2 + z) are parallel, with an interplanar spacing of 3.445(2)Å, and a ring centroid separation of 3.734(3)Å (Table 5). The combination of anion···π and π-π stacking interactions leads the molecules to generate a supramolecular assembly in (011) plane (Figure 4).

**Table 4.** Parameters (Å, °) for Y–X···π interactions in compound (**1**).

| Y–X···Cg(I) | X···Cg | Y···Cg | Y–X···Cg | X-Perp | Symmetry Operation |
|---|---|---|---|---|---|
| C (11)–O(5)···Cg(4) | 3.498(6) | 4.148(3) | 108 | 3.485 | 1 − x, −y, 1 − z |

Cg(4) is the centroid of the rings generated through the atoms (N4A/C8A–C12A).

**Table 5.** Parameters (Å, °) for π-stacking interactions.

| rings (*i*)···(*j*) | Rc | R1v | R2v | α | β | γ | Symmetry |
|---|---|---|---|---|---|---|---|
| *Compound (1)* | | | | | | | |
| Cg(1)···Cg(3) | 3.734(3) | 3.445(2) | 3.530(3) | 3.80 | 19.00 | 22.69 | x, 1/2 − y, −1/2+z |
| *Compound (2)* | | | | | | | |
| Cg(1)···Cg(2) | 3.783(3) | 3.666(2) | 3.773(2) | 15.80 | 4.09 | 14.27 | −1 + x, y, z |
| Cg(2)···Cg(2) | 3.771(3) | 3.391(2) | 3.391(2) | 0.00 | 25.95 | 25.95 | 1 − x, 1 − y, 1 − z |
| *Compound (3)* | | | | | | | |
| Cg(1)···Cg(1) | 4.113(2) | 3.279(2) | 3.279(2) | 0.0 | 37.12 | 37.12 | 2 − x, −y, −z |
| Cg(2)···Cg(2) | 4.218(2) | 3.407(2) | 3.407(2) | 0.0 | 36.14 | 36.14 | 5/2 − x, 1/2 − y, 1 − z |
| Cg(2)···Cg(3) | 4.105(3) | 2.846(2) | 3.949(2) | 31.8 | 15.86 | 46.11 | 5/2 − x, 1/2 + y, 1/2 − z |

Compound (**1**): Cg(1) and Cg(3) are the centroids of the (N1/C1–C5) and (N4/C8–C12) rings respectively.
Compound (**2**): Cg(1) and Cg(2) are the centroids of the (N3/C9–C13) and (N4/C14–C18) rings respectively.
Compound (**3**): Cg(1), Cg(2) and Cg(3) are the centroids of the (N1/C2–C6), (N6/C22–C26), and (C8–C13) rings respectively.

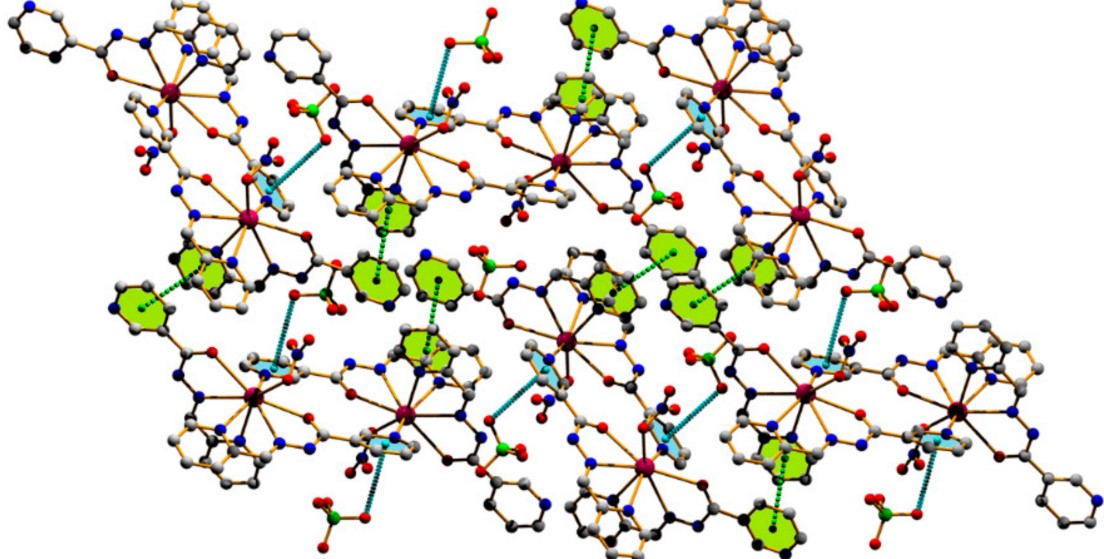

**Figure 4.** Supramolecular network of (**1**) in (011) plane through anion···π and π-π interactions.

The ORTEP view of compound (**2**) is shown in Figure 5 with partial atom numbering. The compound (**2**) crystallized in space group P-1 in which the asymmetric unit consists of half molecular moiety. The full moiety is generated by the symmetry operation of an inversion center. The Pb atom in the asymmetric unit is located on the inversion center at $(0, 0, \frac{1}{2})$ and possesses an octahedral coordination whose equatorial planes are generated by one oxygen and one nitrogen atoms from the ligand and their symmetry related counterparts of another ligand. Two iodine atoms (I1 and I1*. * = −x, 2 − y, 1 − z) occupy the *trans* axial positions, consequently forming a $PbO_2N_2I_2$ chromophore. The apical Pb-I bond length is much longer in comparison to the equatorial bond lengths. A second nitrogen atom N(3) from the ligand lies at a longer distance (2.917Å) than the equatorial Pb(1)–N(1) bond-length (Table 2). In this complex both Pb(1)–N(1) and Pb(1)–N(3) are significantly longer than $\Sigma R_{cov}$. values and closer to $\Sigma R_{vdw}$ values, and thus they can be also considered as tetrel bonds.

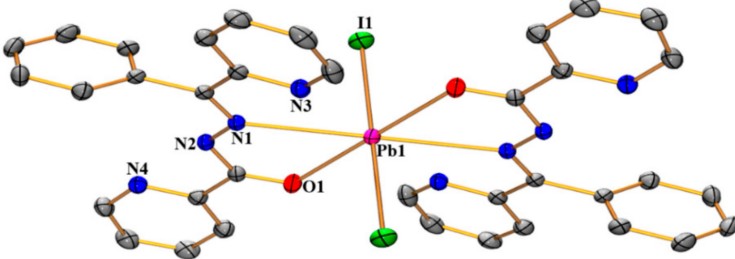

**Figure 5.** Molecular view of compound (**2**) with partial atom numbering. The unlabeled counterpart is produced through the symmetry (−x, 2 − y, 1 − z). Thermal ellipsoids have been generated at 30% probability.

Due to the self-complementary nature of this complex, the pyridine rings (N3/C9–C13) and (N4/C14–C18) are juxtaposed through π-π stacking interaction (Table 5) with a ring centroid distance of 3.783(3)Å, and with an interplanar separation of 3.666(2)Å. The pyridine rings (N4/C14–C18) of the molecules at (x, y, z) and (1-x, 1-y, 1-z) are arranged antiparallel (Table 5), with an interplanar separation of 3.391Å, with ring centroid distance of 3.771(3)Å, and with a ring offset of 1.65Å. The mixture of π-π stacking interaction generates the two-dimensional supramolecular-layered assembly shown in Figure 6.

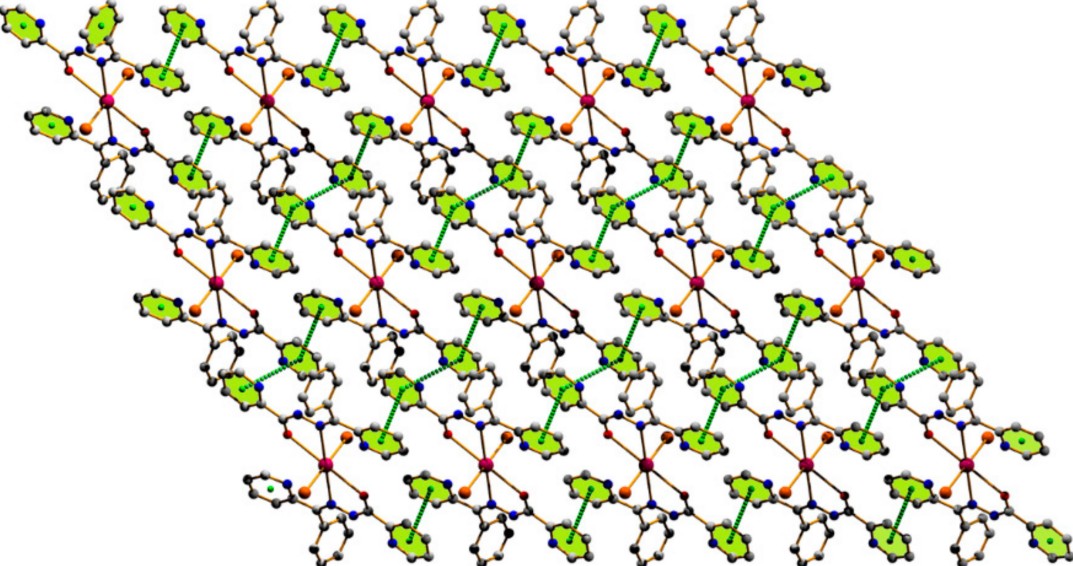

**Figure 6.** Supramolecular self-assembled network in (**2**) in (110) plane through the mixture of π-π stacking forces.

The molecular view of compound (**3**) is included in Figure 7. Two oxygen and two nitrogen atoms of the ligand binds the metal atom and the symmetry (2 − x, y, 1/2 − z) related counterpart of another ligand occupies the equatorial plane and one water molecule occupies the axial position, thus generating a $PbO_5N_4$ chromophore. The Pb–O bond distances range from 2.581(3) to 2.737(7)Å while the Pb–N bond lengths range from 2.675(3) to 2.701(3)Å (Table 2). In the solid state, the compound (**3**) is stabilized through O–H···O, C–H···O and π–π stacking forces (Tables 3 and 5). In (**3**), two perchlorate oxygen atoms O5 and O6 act as acceptor to the water oxygen atom O7 and carbon atom C25 of the pyridine ring; thus forming a $R_4^4(26)$ dimeric ring (Figure S2). In another substructure, the pyridine and aryl rings (N6/C22–C26) and (C8–C13) in the molecules at (x, y, z) and (5/2 − x, 1/2 + y, 1/2 − z) are juxtaposed through weak π–π stacking interaction with an intercentroid distance of 4.105(3)Å. The pyridine ring (N6/C22–C26) further interacts with another (N6/C22–C26) ring of the partner molecule; thus leading the molecules to propagate along (010) direction. The pyridine rings at (x, y, z) and (2 − x, −y, −z)

are antiparallel, with an interplanar separation distance of 3.279(2)Å, and distance between ring centroids of 4.113(2)Å, with a ring offset of 2.48Å (Table 5). The multi-π stacking lead a supramolecular self-assembled framework in (011) plane (Figure 8).

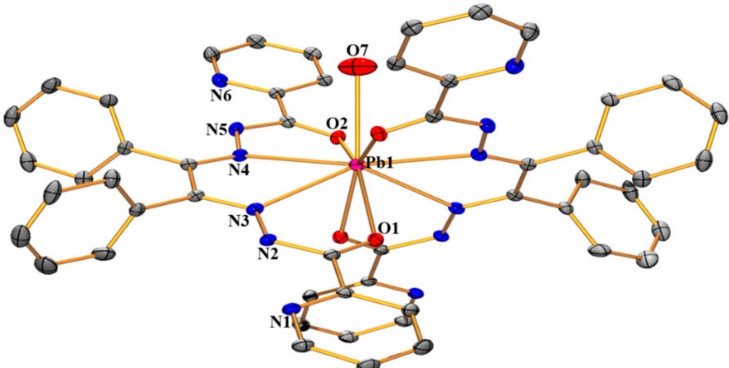

**Figure 7.** Molecular view of compound (**3**) with partial atom numbering. The unlabeled counterpart is generated through the symmetry operation (2 − x, y, 1/2 − z). Ellipsoids have been generated at the 30% probability.

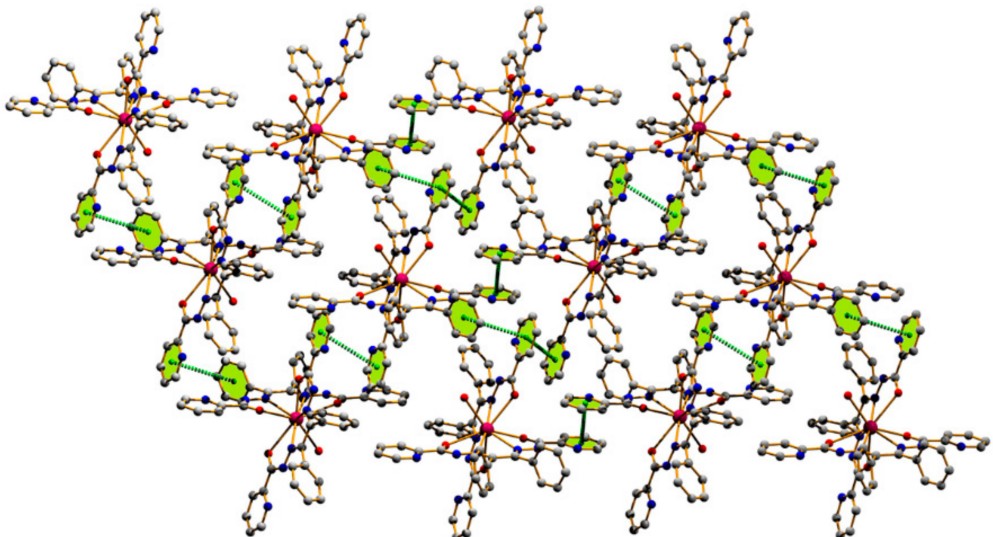

**Figure 8.** Self-assembled supramolecular network in (**3**) through π-π stacking in (011) plane.

### 3.2. Hirshfeld Surface Analysis

Following the pattern of the solid-state structure of all compounds, we were interested to quantify the noncovalent interactions. In this investigation, the contacts responsible in building supramolecular assemblies are evaluated. The Hirshfeld surface analysis of the investigating compounds are performed and mapped over $d_{norm}$ (Figure 9, left column) in the range (−0.474 to 1.891Å), (−0.471 to 1.154Å), and (−1.295 to 1.522Å) for (**1**), (**2**) and (**3**) respectively. The shape-index plot of all three compounds are shown in Figure 9 (middle column). The scattered points of the fingerprint plots (Figure 9, right column) evidence all contacts that are exhibited by the structures. The symmetry-generated counterpart of the structures were not considered for this calculation. Therefore, the large depression on the metal centers and the bridging N/O atoms are due to the symmetry generated counterpart of the structure. For instance, the Pb···N/N···Pb interaction is evidenced by dual discrete spikes in the breakdown fingerprint plot (Figure S3) of compound (**1**) since one nitrogen atom binds with another Pb atom through the inversion center whereas for compounds (**2**) and (**3**), only one spike is evident (Figures S4 and S5). That means there exists both Pb···N (0.7%) and N···Pb (0.7%) interaction in compound (**1**) but for compounds (**2**) and (**3**), only Pb···N interaction exists. There is no signature of N/O···Pb interactions

in compounds (**2**) and (**3**). Due to the variety of the bridging mode, the contribution are different and the Pb⋯N/N⋯Pb interaction comprises 1.4%, 2.7%, and 1.4% of the total Hirshfeld surface area of compound (**1**), (**2**), and (**3**) respectively. The I⋯H/H⋯I contacts in compound (**2**) are evidenced by two distinct spikes on the fingerprint plot, where I⋯H interaction contributed more (12.1%) compare to the H⋯I counterpart (4.7%). It is clearly evident that the spikes in the breakdown fingerprint plots correspond to the O⋯H/H⋯O interactions that are very piercing and distinct for compound (**1**) (Figure S3) whereas no such distinct spikes are evidenced (Figure S4) for compound (**2**) and compound (**3**) exhibits one sharp spike (Figure S5). The spikes in different regions of the fingerprint plot comprises 43.5% in (**1**), only 5.6% in (**2**), and 25.7% in (**3**). The N⋯H/H⋯N contribution is also higher in compound (**1**) compare to compounds (**2**) and (**3**) (see Figures S3–S5). In all the structures, the N⋯H contact contributed more than it's H⋯N counterpart. Similarly, we have analyzed and quantified all contacts that are exhibited by the structures are included in Figures S3–S5. Following the structural description, we have analyzed π-π contacts by inspecting shape-index surfaces (Figure 9). The adjacent red and blue triangles on the shape-index surface that are marked by the red circle are characteristic of π-π stacking contacts [56,57]. The π-π interaction contributed 4.6% in (**1**), 5.9% in (**2**), and 2.6% in (**3**) that are evidenced by breakdown fingerprint plots. The H⋯H contacts comprised of 16.0% in (**1**), 40.1% in (**2**), and 34.4% in (**3**) that are shown in Figures S3–S5).

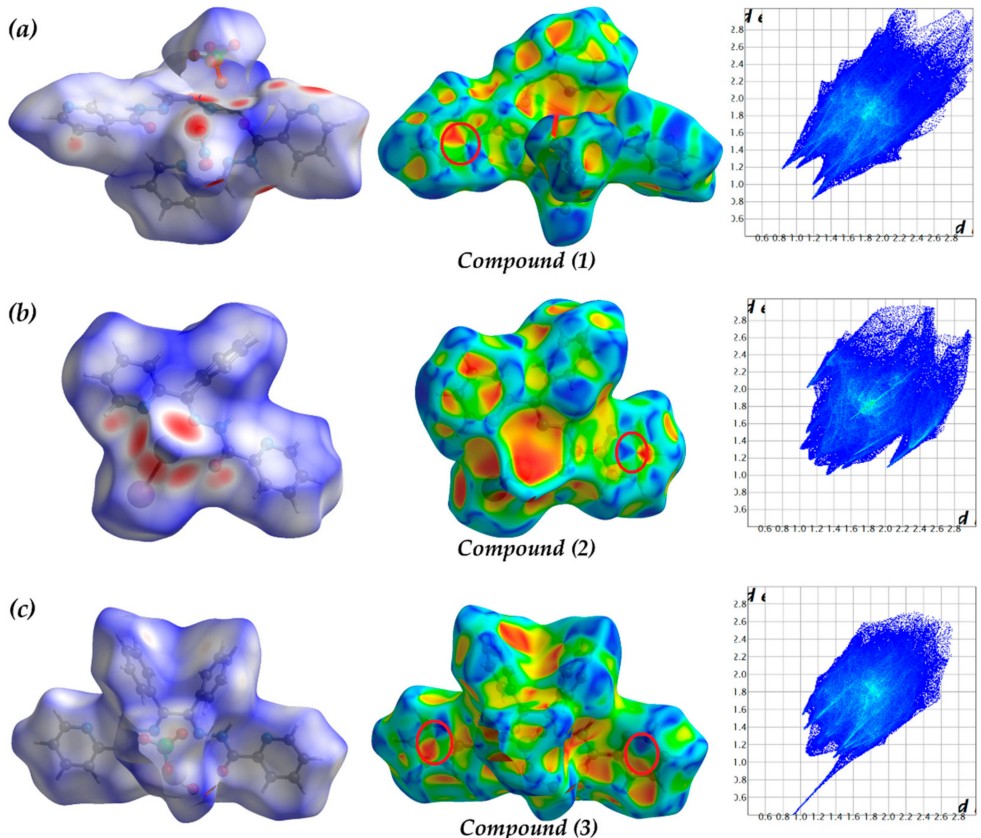

**Figure 9.** Hirshfeld surface, shape-index surface and fingerprint plot of compounds (**1**) (**a**), (**2**) (**b**), and (**3**) (**c**) that are mapped with $d_{norm}$ (left column), *shape-index* (middle column) and full fingerprint plot (right column).

### 3.3. Theoretical Study

This study is intended to analyze the π–π and anion–π interactions observed in the crystal packing of the Pb(II) compounds.

For compound **1**, we have first analyzed the anion–π interaction described above (see Figure 4). A careful inspection of the structure shows that the interaction of perchlorate anion with the ligand is

ditopic. That is, one O-atom is over the coordinated pyridine ring of the nicotine moiety and another one with shortest distance (3.08 Å, see Figure 10) is located over the C atom of the hydrazido group (π-hole interaction) [58,59]. This C atom is a good π-hole donor [60] due to the adjacent Pb–O coordination bond that enhances the π-acidity of C. We have computed the interaction energy of the model shown in Figure 10a, which is very large ($\Delta E_1 = -173.9$ kcal/mol) because of the significant electrostatic attraction between the dicationic $[Pb_2(L^1)_4(NO_3)_2]^{2+}$ moiety and the anions. To evaluate the importance of the O$\cdots$π-hole contact we have utilized a model where chlorate anion instead of perchlorate was used (see Figure 10b). By doing so, the interaction energy weakens to $\Delta E_2 = -156.1$ kcal/mol and, consequently, the contribution of each O$\cdots$π-hole interaction can be estimated by difference, which is $-8.9$ kcal/mol $[\frac{1}{2}*(\Delta E_1 - \Delta E_2)]$. This interaction energy is consistent with other π-hole interactions [60].

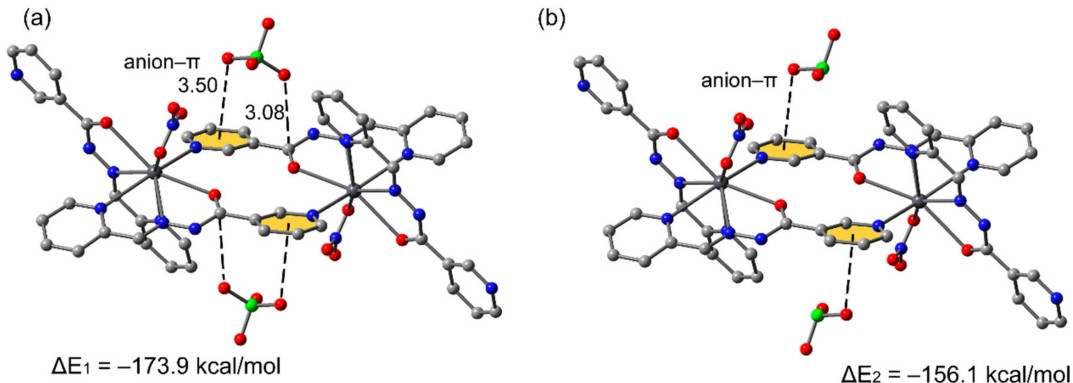

**Figure 10.** (**a**,**b**) Theoretical models of compound (**1**) (distances in Å). H-atoms are not shown for clarity.

In complex (**2**) the π–π interactions previously described in Figure 6 have been studied, which are crucial in the crystal packing of this complex. In Figure 11, we show the self-assembled dimer used to evaluate the interaction energetically. The resulting binding energy is strong ($\Delta E_3 = -26.5$ kcal/mol) due to the antiparallel arrangement of the pyridine rings and the coordination of the ligand to the Pb(II) that enhances the dipole-dipole interaction.

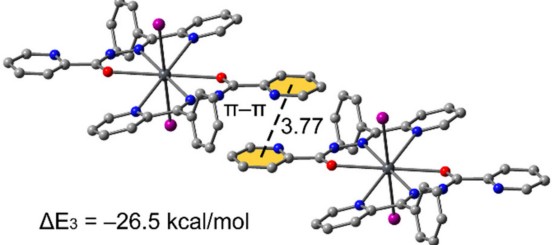

**Figure 11.** Theoretical models of the dimer of compound (**2**) (distance in Å). H-atoms omitted for clarity.

Finally, in compound (**3**) we have studied the intricate combination of interactions that are established between the perchlorate anion and the complex (see Figure 12). The four O-atoms of the anion are involved in several contacts, which are OH$\cdots$O H-bonding interactions with the coordinated $H_2O$ ligand, CH$\cdots$O H-bonds with the aromatic H-atoms and finally, an interesting O$\cdots$chelate ring (CR) interaction. The latter interaction has been previously described in the literature, where the chelate rings are able to interact favorably with anions or lone pair donor atoms [61]. We have computed the binding energy for the interaction of one perclorate (we have considered the system as binary, $[Pb(L^3)_2(H_2O)][(ClO_4)]\cdots[(ClO_4)]$), which is very large due to the pure electrostatic attraction between the positive Pb(II) complex and the anion, $\Delta E_4 = -79.5$ kcal/mol. To evaluate the strength of the OH$\cdots$O H-bond, another model has been computed where the coordinated water molecule has been eliminated. Therefore, the H-bond is not formed and the interaction energy weakens to $\Delta E_5 = -72.1$ kcal/mol and, consequently the H-bond contributes in $-7.4$ kcal/mol.

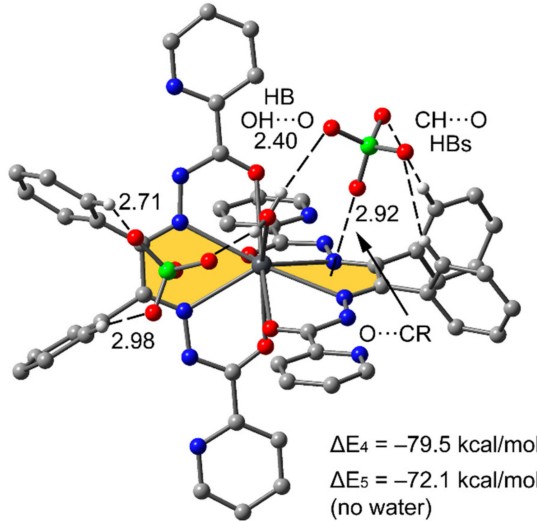

**Figure 12.** Theoretical model of compound (**3**) (distances in Å). H-atoms omitted for clarity apart from those participating in H-bonds.

## 4. Conclusions

The synthetic protocol to obtain three new Pb(II) coordination compounds involving nicotinhydrazido and picolinhydrazido- ligands and I$^-$ and ClO$_4$$^-$ counter anions is reported in this manuscript. The complexes were characterized in the solid state by X-ray diffraction. In compound (**1**), the perchlorate interacts with the organic ligands via ditopic anion–π and O···π-hole interactions. These interactions have been analyzed theoretically using the DFT-D method. Moreover, in compound (**3**), the anions interact with the Pb-complex through a network of OH···O and CH···O H-bonding interactions and also unconventional anion···CR (chelate ring interactions). Finally, in compound (**2**), we have studied the strong and antiparallel pyridine–pyridine stacking assembly that is reinforced by their coordination to Pb(II).

**Supplementary Materials:** The following are available online at http://www.mdpi.com/2073-4352/9/6/323/s1, Figures S1–S5: Packing diagrams and decomposed fingerprint plots of the title compounds.

**Author Contributions:** G.M., F.I.Z. synthesized the compounds. S.K.S. performed the Structural and Hirshfeld surface analysis. E.L.T., A.B., V.S. formal analysis. A.F. performed the D.F.T. study. G.M., S.K.S. and A.F. planned the research. G.M., S.K.S. and A.F. wrote the manuscript. All authors revised the manuscript.

**Funding:** The publication was prepared with the support of the RUDN University Program 5-100. This research was funded by MINECO/AEI from Spain, project number CTQ2017-85821-R FEDER funds.

**Acknowledgments:** We acknowledge the support provided by R. Frontera from the Centre de Tecnologies de la Informació (CTI) at the UIB for computational facilities.

**Conflicts of Interest:** The authors declare no conflict of interest.

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
