# Peer review of "Supramolecular Assemblies in Pb(II) Complexes with Hydrazido-Based Ligands"

_crystals, doi:10.3390/cryst9060323_

Round 1
Reviewer 1 Report
Please find my annotation (major and minor ones) in the atttached pdf-file.
Some general comments:
The list of references is quite long (which is not a problem a-priori), including some wrong ones and a considerable number of self-citations.
Some details of the structural disussion should be improved.
The NCI method seem to be inappropriate as it provides significantly different values for Pb-O and Pb-N bonds!

Author Response
Responses to Reviewer 1.
First of all we would like to thank this reviewer for his/her careful reading of the manuscript, suggestions and corrections. We have modified the manuscript accordingly, as detailed below:
Please find my annotation (major and minor ones) in the atttached pdf-file.
Some general comments:
The list of references is quite long (which is not a problem a-priori), including some wrong ones and a considerable number of self-citations.
Some details of the structural disussion should be improved.
The NCI method seem to be inappropriate as it provides significantly different values for Pb-O and Pb-N bonds!
Reply: We have modified the references and improved the structural discussion part following each and every comment. The NCI plot calculations have been eliminated.
- Line no 25: I’m strongly against using the word “center” in general for the central ion of a coordination compound. In many cases, the metal ion is neither in the center of the complex nor of the structure.
Reply: We have eliminated the word “center”
- Line no 28: This sentence is not true in general. It is difficult if interatomic distances (from structure analyses) are considered, but there are (DFT, e.g.) tools to analyze the chemical bond.
Reply: This sentence has been eliminated
- Line no 41: What do the authors mean by increasingly used? The number of cars worldwide which use lead start batteries?
Reply: This sentence has been removed.
- Line no 47: Just judging by the titles of references 16-21: they have nothing to do with Pb(II)! Please check and correct!
Reply: We are sorry for this important mistake. These references have been either removed or updated.
- In page 2, Scheme 1: These images imply that the lone pair of Pb(II) is sometimes "located" in an p-type orbital (a) and sometimes in an s-type orbital (b). Is that a realistic scenario? How large is the energy difference between these two hybridizations? Moreover, it is irrelevant in further discussions in the text.
Reply: Since it is irrelevant, we have eliminated Scheme 1.
- Line no 89: What is Ost and CCst?
Reply: “St” stands for stretching. We have clarified it.
- Line no 119: Please provide an explanation for the color codes used.
Reply: The detailed explanation of the used color schemes of Hirshfeld surface is included in the revised manuscript.
- Line no 126: ref 59 deals with codes for transition metal atoms Sc-Hg!
Reply: Thank you for taking this to our attention. Ref 59 has been corrected.
- Line no 130: Please provide a color scale bar in the respective figures.
Reply: Since the reviewer has advised that the NCIplot is inappropriate, we have eliminated the NCIplot results entirely from the manuscript
- In page 5, Table 1: It makes no sense to state 4 decimal digits if the third one is already "uncertain" within the given standard deviations. Please reduce the number of digits to the relevant ones.
Reply: Fixed
- Line nos. 147-149: ...occupies the seventh coordination site with a distance of .... This is wrong. N4A is not in an inversion center. ALL atoms of (1) are on a general site!
Reply: Fixed.
- Line no 166: What does that mean "the first substructure" ?
Reply: Omitted and revised.
- Line no 169: ?
Reply: Omitted and revised.
- Line no 171: The dimeric ring does not lead the molecules.
Reply: Fixed.
- In page 7, Table 3: in Å /°. symmetry operation
Reply: Fixed.
- Line no 179: What is a/e "final sub-structure"?
Reply: Fixed.
- Line no 208: The defining Pb atom in the asymmetric unit is located on the inversion center at 0, 0, 1/2 (although the equvalent position 0, 1, 1/2 is stated in the cif) -x, 2-y, 1-z is a symmetry operation but not a position!
Reply: Fixed.
- Line no 222: The pyridine rings are strictly parallel or not?
Reply: We have changes strictly parallel by antiparallel.
- Line no 236: ?
Reply: Revised.
- In page 10, line no 240: How large is the contribution of pi-pi-interactions at distances of > 4.1 Å? I expect it is neglectable.
Reply: We agree with the learned reviewer, the strength of the interaction is weak due to longer distance but have taken following the directionality of the force.
- Line no 262: What does that mean for molecules (2) and (3): Only one half (the defining one in the asymmetric unit?) of the molecules were used for calculation?
Reply: Since the symmetry generated counterpart comprises the same and duplicate contribution of contacts on the surfaces, we have omitted that part only. Yes, only one half of the asymmetric unit were used for the calculations.
- Line no 268: Interactions with an overall contribution of less than 1%? What's the relevance?
Reply: The Pb···N/N···Pb interaction comprises total 1.4% to the total Hirshfeld surface of the molecule. Moreover, our intention is to quantify each individual interaction in a novel visual manner.
- Line no 283: Where are red and blue triangles in (3)?
Reply: We have marked the red and blue triangles on the shape-index surface by red encircles) (Figure 9, middle column).
- In page 12, Figure 9 caption: Provide a color-code scale bar!
Reply: No color-code scale bar is provided by the used CrystalExplorer software.
- Line nos. 303-307: As far as I can see, the perchlorate anion is also involved in multiple H-bonds. If you replace ClO4(-) by ClO3(-), the number of H-bonds will also be affected. How did you make sure, that no contributions of missing H-bonds contribute to delta E(2)?
Reply: Since we are only interesting in evaluating the contribution of the O···p-hole interaction, the utilization of these two theoretical models is adequate (only for this particular purpose). Of course the perchlorate anion is establishing interactions with neighbor molecules in the solid state that would be affected in the ClO4(-) is replaced by ClO3(-).
- In page 13, Figure 10 caption: What is s? Provide a color-code scale bar
Reply: The NCIplot has been eliminated
- Line 318: What is s?
Reply: s is the reduced density gradient
- Line nos. 326-328: The finding of non-covalent Pb-N bonds makes no sense and sheds a very bad light on the following results which are discussed in great detail.
I'm very much concerned that the method is not useful/applicable here. And is is not justified to pick just those results which fit into the discussion of the manuscript.
Reply: Since this referee is concerned about the reliability of this method, we have eliminated the NCIplot results
- In page 14, Figure 11 caption, Line no 342: color code
Reply: The NCIplot has been eliminated
- In page 14, Figure 11 caption, Line no 343: On page 10, this assembly was discussed as "strictly parallel".
Reply: We have changed strictly parallel by antiparallel
- Line no 359: The distance does not justifiy any color. The color comes from some s value which is undefined/unexplained in the manuscript.
Reply: The NCIplot has been eliminated
- Line no 369: I see IR as the only spectroscopic method.
Reply: Corrected
Reviewer 2 Report
Three new Pb(II) complexes using Schiff base hydrazido-based ligands and different counterions were synthesized and well characterized by single crystal X-ray diffraction and standard spectroscopic methods. Furthermore Hirshfeld surface analysis and Theoretical study were carried out.
In my opinion this manuscript should be accepted with minor revisions and the authors should correct the following:
In Line 82-83: For NaClO4, first the amount of g (letter g is missing) and then the mmol
In the experimental part: the author use some abbreviations such as st, oop, …but the meanings don’t appear in the paper. Furthermore the abbreviation st, sometimes is near the text and sometimes there is a space. Please check it.
In Table 1: For the complex 3, Monoclinic should be start by capital letter.
Line 156: [Pb(L1)2][NO3]·[ClO4], author must change the ] by ] and use · in the complex but not in other complexes. Please check it.
Author Response
Responses to Reviewer 2.
First of all, we would like to thank this reviewer for his/her careful reading of the manuscript, suggestions and corrections. We have modified the manuscript accordingly, as detailed below:
Three new Pb(II) complexes using Schiff base hydrazido-based ligands and different counterions were synthesized and well characterized by single crystal X-ray diffraction and standard spectroscopic methods. Furthermore Hirshfeld surface analysis and Theoretical study were carried out.
In my opinion this manuscript should be accepted with minor revisions and the authors should correct the following.
Reply: Thank you for supporting publication.
In Line 82-83: For NaClO4, first the amount of g (letter g is missing) and then the mmol
Reply: Fixed
In the experimental part: the author use some abbreviations such as st, oop, …but the meanings don’t appear in the paper. Furthermore the abbreviation st, sometimes is near the text and sometimes there is a space. Please check it.
Reply: Fixed
In Table 1: For the complex 3, Monoclinic should be start by capital letter.
Reply: Fixed.
Line 156: [Pb(L1)2][NO3]·[ClO4], author must change the ] by ] and use · in the complex but not in other complexes. Please check it.
Reply: Fixed